# The Immediate Hypoalgesic Effect of Low and High Force Thoracic Mobilizations in Asymptomatic Subjects as Measured by Pain Pressure Thresholds (PPT)

**DOI:** 10.3390/diagnostics13030544

**Published:** 2023-02-02

**Authors:** Charilaos Syrgiamiotis, Georgios Krekoukias, Katerina Gkouzioti, Clair Hebron

**Affiliations:** 1Faculty of Health, School of Health Professions, University of Brighton, 49 Darley Road, Eastbourne BN20 7UR, UK; 2Laboratory of Advanced Physiotherapy, Physiotherapy Department, School of Health & Care Sciences, University of West Attica (UNIWA), 12243 Athens, Greece

**Keywords:** thoracic mobilization, mobilization force, hypoalgesic response, pressure pain thresholds

## Abstract

Physiotherapists commonly use mobilizations for treating patients with thoracic spine pain (TSP). There is evidence to suggest that spinal mobilizations can decrease pain. Different doses of mobilization treatment are applied, however there is a paucity of evidence on the influence of these dosage parameters. The effect of different forces of treatment remains unknown. This study aimed to investigate whether there was a difference in the hypoalgesic effect of high and low force thoracic mobilizations. This single-blinded, randomized, within-subject, repeated measures, cross-over design recruited 28 asymptomatic participants. Participants received the experimental conditions of high (200 N) and low force (30 N) mobilizations to T6 at least 48 h apart. Pressure pain thresholds (PPTs) were measured before and immediately after each experimental intervention at three different standardized sites. The results demonstrated that high force thoracic mobilizations caused a significant increase in PPT measures compared to low force mobilizations. This effect was detected at all PPT sites. This study suggests that high force thoracic PA mobilizations cause a significantly greater hypoalgesic response in asymptomatic participants than low force thoracic mobilizations. The hypoalgesic response seems to be elicited not only locally at the site of the intervention, but in a widespread manner.

## 1. Introduction

Spinal pain is a ubiquitous problem affecting a large proportion of the population [1]. Low back pain is the most common musculoskeletal complaint [2], with a lifetime prevalence of 57% [1,3], while thoracic spine pain has a lower prevalence of approximately 13–17% [3,4]. Physiotherapists use a range of modalities to treat spinal pain, with national guidelines advocating, exercise with or without psychological support and manual therapy [3,4].

Spinal passive joint mobilizations are manual therapy techniques commonly used by physiotherapists in the management of musculoskeletal conditions of the thoracic spine [5,6] with the aim of decreasing pain, stiffness, and muscle activity, and increasing the range of movement [7,8,9,10]. Mobilization consists of low velocity passive oscillatory movements within or at the limit of the joint’s range of motion [5]. It has been proposed that the effects of mobilizations are predominantly neurophysiological [11]. Spinal and supraspinal neurophysiological mechanisms have been proposed to be involved in mobilization induced analgesia [8,11,12]. Pain modulation at the level of the spinal cord is supported by studies reporting an immediate reduction in temporal summation following spinal manual therapy [13,14,15]. Mobilizations may also reduce central sensitization through the depression of dorsal horn neurons [16].

In addition, there is evidence to suggest that a descending pain inhibition mechanism, activated by the central nervous system (periaqueductal gray) after the application of mobilizations, may be responsible for providing a hypoalgesic response in a widespread manner [12]. Skyba et al. [17] provided support for involvement of the PAG using pharmacological manipulation of neurotransmitters, and studies using functional magnetic resonance reported a trend towards decreased activation of the brain areas associated with pain [18] and changes in the functional connectivity in supraspinal areas [19] following mobilization.

A number of randomized controlled trials (RCTs) have demonstrated that spinal mobilizations have a hypoalgesic effect, as measured by pain pressure thresholds (PPT), in asymptomatic participants and similarly in populations suffering from musculoskeletal pain [20,21,22,23,24]. This hypoalgesic response seems to be detected not only locally at the site of mobilizations but also at distant locations in the limbs [20,21,23]. Most of the studies have been conducted on the cervical spine. One controlled, single blinded study investigating the hypoalgesic effect of thoracic mobilizations in asymptomatic participants reported a significant increase in pain pressure thresholds (PPTs) compared to a control group [25]. 

Treatment dose is a term used by clinicians to describe the parameters of the mobilization treatment applied by the therapist [26]. The mobilization treatment dose, as used in clinical practice, consists of the following elements: grade (related to force), amplitude, rate, rhythm, and duration [27]. Despite the wide use of mobilization techniques in clinical practice, there is a paucity of evidence about the optimal dose, and it is not known whether the dose of the treatment influences the hypoalgesic response. A few studies have investigated different dosages of lumbar mobilizations and reported no significant influence on PPTs between different rates and different amplitudes [28,29]. However, Pentelka et al. [30] suggested that a longer treatment duration may have an increased hypoalgesic effect. Recently, another study revealed an overall increase in PPTs after the application of rotatory posteroanterior mobilizations on T4 with different rates, but a statistical analysis did not show any significant difference among PPT measures [31]. Two similar pilot studies have investigated the effects of changing the amount of the force of mobilization in peripheral joints [32,33]. Vicenzino et al. [32] suggested that a lateral glide to the elbow with a force threshold of 62.2 N might be sufficient to produce hypoalgesia in patients with lateral epicondylalgia, while McLean et al. [33] reported that a hypoalgesic response was elicited by a magnitude of force equal to 66% of the maximum force applied (113.2 N). One randomized controlled study [34] compared the application of mobilization with low force (30 N) and high force (90 N) on the cervical spine of symptomatic participants. The results showed no significant differences on PPTs. However, in another recent study Hebron [35] reported that higher treatment forces may be associated with a greater immediate reduction in pain measured by PPT and Verbal Rating of Pain (VRP) after the application of lumbar mobilizations in patients with chronic low back pain. 

The pain relieving effect of different magnitudes of force of mobilizations remains unknown. This study set out to investigate the effect of thoracic mobilizations applied with high forces and low forces on PPTs of asymptomatic participants. The hypoalgesic effect was measured by PPTs at locations close to and distant from the intervention site, so any local or widespread effects were investigated for further insight into the extent of the hypoalgesic effect.

## 2. Materials and Methods

### 2.1. Participants

The power analysis software G-Power (version 3.1, Universitat Kiel, Kiel, Germany) was utilized to determine the a-priori sample size. The effect size of 0.21, an alpha and beta of 0.05, and a power of 0.95 [36], indicated a sample power of 25 participants. To allow for the eventuality of dropouts, we included twenty-eight asymptomatic participants (9 males and 19 females) who were recruited via university email. The participants had a mean age of 29.4 (±1.87) years and a mean body mass index of 23.2 (±0.6) kg/m^2^. Ten were physiotherapy naïve. Participants were excluded from this study if they had a history of spinal pain in the last 12 months or any precautions or contraindications to manual therapy [37]. Participants gave written informed consent prior to taking part in the study. Ethical approval was granted by the University of Brighton (Brighton, UK) ethics panel (2 July 2014).

### 2.2. Research Design and Experimental Procedure

This study used a single-blinded, randomized, within-subject, repeated measures, crossover design. As we included asymptomatic participants, this could be regarded as a pilot study. Each subject attended two experimental sessions, at least 48 h apart, in order to control for carry-over effects. Participants received each experimental condition in a randomized order. On the first attendance, each participant was asked to choose one of the two small wrapped pieces of paper. Each paper was assigned the letter “H” or “L”, representing the high force and the low force mobilization respectively. To assess changes to pain level sensitivity, PPTs were measured before and also immediately after each experimental condition, using a digital pressure algometer fitted with a 1 cm tip FPX^®^ (Wagner instrument, Greenwich, CT, USA), which was applied perpendicular to the skin by a research assistant who was blind to the condition applied. Pressure algometry has been used to measure PPTs and shown good to excellent reliability within and across consecutive days [38,39,40].

The participants were instructed to signal to the researcher when they identified that the sensation produced by the algometer ‘changed from pressure to discomfort or pain’. To familiarize participants with the algometer application, a ‘practice PPT’ on a body part not involved in the study was taken before the experimental measurements. Three PPT measurements were taken before and immediately after each experimental procedure, resulting in a total of six measurements (three before and three after) at each site. Three sites were chosen in order to establish the local or widespread hypoalgesic response (Figure 1). These sites were marked to standardize the repositioning of the algometer. 

Then, a physiotherapist with 7 years of experience on musculoskeletal conditions and manual therapy techniques, applied the corresponding chosen experimental intervention. Oscillatory, grade III, central PA mobilizations were applied to the T6 spinous process using a pisiform grip [5]. A metronome set to 60 beats per minute (1 Hz) was used to control the rhythm of mobilization (Seiko DM-51, Seiko Instruments Inc., Tokyo, Japan). The mobilization was applied in four sets in each experimental intervention with a 1 min duration and a 1 min rest period between sets. The mobilizations were performed by the same experienced manual therapist. The experimental conditions consisted of either:(1)High force mobilization, target peak force 200 N(2)Low force mobilization, target peak force 30 N

The force of mobilization was measured and monitored by the use of a plinth mounted on a force plate (AMTI OR6-7 Advanced mechanical Technology Inc., Watertown, MA, USA) linked to a computer screen, so the researcher could gain real time feedback of the mobilization being applied. This instrumentation has been utilized in other studies [7,28,29,30,41]. The order of the experimental procedure is demonstrated in Figure 2.

### 2.3. Data Analysis

All analyses were done using the statistical package for social science (SPSS) software (version 22, SPSS Inc., Chicago, IL, USA). All the data were tested for normality using the Shapiro–Wilk test. A three-way repeated measures (ANOVA) was utilized to test PPT data with three within participants’ variables, condition (two levels: high force and low force), time (two levels: before and after), and site (three levels: T6, mid-forearm and fibula). Agresti and Finlay [42] suggested that analysis of variance (ANOVA) is robust to be used even when minor departures from normality remain. 

The Intraclass Correlation Coefficients (ICC) were used to test the reliability of the baseline PPT measurements for the two experimental conditions (single measure, two way mixed). The standard error of measurement (SEM) and the minimal detectable change (MDC) were also calculated.

A chi-square analysis was used to test for a statistically significant relationship between the magnitude of force and participants’ response measured by changes in PPT.

The mean peak forces were calculated using a Macro written in Visual BASIC for Applications in Microsoft Office Excel 2007 software (Microsoft Inc. Redmond, WA, USA). This software calculated the mean high or low force of each applied set of mobilization. 

The cumulative proportion of the responders’ analysis was used as a method to describe the likelihood of response over a range of response levels [43].

## 3. Results

All 28 participants completed the study successfully, with no adverse effects. 

The chi-square analysis based on changes in PPT of SEM or greater using the frequency of responders at each PPT site for each experimental intervention, found that there was a significant association between the magnitude of force and whether participants responded immediately after each intervention (Table 1). 

The chi-square analysis using the frequency of responders exceeding MDC, revealed that there was no association between magnitude of force and participants’ response, at least for the T6 site and fibula site. A significant association appeared to exist only for the mid-forearm site (Table 2). 

Cohen’s D was calculated to show the effect size in each site [44]. For the T6 site, Cohen’s D is 0.3, for the mid-forearm site it is 0.2, and for the fibula site it is 0.0.

### 3.1. Reliability of Baseline Data

The reliability statistics demonstrated good-excellent between-day, intra-rater reliability at all measurements sites (Table 3).

The mean peak high force recorded in this study was 196.3 N (±21.01) and the mean peak low force was 31.8 N (±3.1).

The consistency of the different magnitude of force for each experimental condition was recorded via the force platform. Examples can be seen in Figure 3. 

### 3.2. Cumulative Responders Analysis

The cumulative proportion of the responders analysis (Figure 4) demonstrated that at the T6 paravertebral site, approximately 85% of the participants experienced an increase in PPTs following the high force mobilization. The corresponding proportion of responders after the low force mobilization was approximately 65%. Furthermore, according to the graph, the level of response was greater after the high force mobilization than the low force.

Figure 5 demonstrates that at the Mid-forearm site the proportion of responders after the high force mobilization was approximately 79%, and after the low force, approximately 69%. Similarly, the level of response was greater immediately after the high force intervention.

The fibula site presented the minimum difference regarding the proportion of responders (Figure 6). Approximately 79% of participants responded after high force mobilization and approximately 71% after low force. However, in agreement with the other sites high force intervention caused a greater level of response compared with the low force. 

### 3.3. Main Analysis

The changes in PPT sites before and after each experimental condition, and the actual change in PPTs for each site, are depicted in Figure 7 and Figure 8, respectively. The mean baseline values, the mean increase (kg/cm^2^), and the mean percentage change (%) of each PPT site after each experimental condition are displayed in Table 4.

The research question in this study was to establish the effect of thoracic mobilizations applied with high and low forces. This question is answered by time × condition interaction effect. The ANOVA found a significant time × condition interaction effect, indicating that there was a significant difference in the change in PPT between the high and low force mobilization, with the high force mobilizations eliciting a significantly greater increase in PPT compared to the low force mobilizations. Specifically, following the high force mobilization, at T6 level there was a 25% (±29.5) change in PPT after the high force mobilization, at mid-forearm a 19.7% (±24) change, and at fibula a 14.6% (±19.5) change. The percentage changes after the low force mobilization were 4.4% (±16.8) for T6 level, 6.8% (±14.1) for mid-forearm, and 6.2% (±13.3) for the fibula site. 

The time × site interaction effect failed to reach significance (F_1,28_ = 0.812, *p* = 0.449), suggesting that there was no difference in the pre-post mobilization change in PPT at the different measurement sites. 

## 4. Discussion

The primary findings of this study revealed that higher force thoracic PA mobilizations on asymptomatic volunteers elicited a significantly greater immediate hypoalgesic response than lower force thoracic mobilizations (time × condition interaction effect). The effect size for the difference between high and low mobilizations was small. When investigating the difference between different mobilization treatment doses, a small effect size is anticipated, as both doses would be expected to produce hypoalgesia via both specific and non-specific effects. 

The majority of studies investigating the effects of mobilizations use Grade III amplitude of mobilizations [20,21,22,23,45]. However, no details about the magnitude of force applied are given in these research papers, so comparisons cannot be made. One study comparing high force (90 N) with low force (30 N) cervical mobilizations in patients with neck pain found no significant differences on PPTs (measured at three different sites) [34]. However, this study was conducted in a different area of the spine, and on a symptomatic population, where differences in response may be magnified due to factors such as severity, irritability of the complaint, and patients’ beliefs and fears. In addition, the difference in our findings to that study might be due to the difference in treatment duration, as the current study employed a longer treatment duration, and the difference between forces may not have been evident with a shorter treatment duration. The findings of this study are in agreement with a randomized controlled trial using chronic low back pain patients, which reported that force of mobilization treatment had a significant mediating effect on PPT and verbal rating pain on movement, with greater forces of mobilizations creating a greater analgesic effect [35]. The findings of this study provide further evidence that higher forces of treatment may create a greater hypoalgesic response. Clinically, these findings suggest that, where pain and patients’ beliefs allow, clinicians should consider using higher treatment forces.

The hypoalgesic effect was significantly greater after the high force mobilizations not only locally at the site of the intervention, but at all sites, suggesting a widespread effect (see Table 3). However, the effect size diminished at more distant sites. Two systematic reviews stated that spinal manual therapy seems to induce a widespread analgesic effect in healthy participants, in participants subjected to experimentally induced pain, and in patients with musculoskeletal pain [9,46]. Similarly, Sterling et al. [23] applied cervical mobilizations on patients with whiplash associated disorders and found increased PPTs on the treatment group locally and distantly from the site of mobilization (24.1% at the cervical spine, 11.3% at the elbow, 7.8% at the tibialis anterior on the leg). In the current study, the statistical analysis showed that the time*site interaction effect was not significant, suggesting that there were no significant changes between the PPT sites. The overall analysis supports a widespread hypoalgesic effect of mobilizations. The decreased effect sizes suggest that the hypoalgesic effect is diminished at more remote locations. The effect sizes are small, which is to be expected as this study was comparing two similar interventions each of which induce a hypoalgesic response. Larger effect sizes might be anticipated when comparing a treatment to a control intervention. 

It is hypothesized that mobilizations may induce analgesia mediated by neurophysiological mechanisms at the spinal cord [13,14] and descending pain inhibition mechanisms [8,12]. The results of this study seem to support the hypothesis that this widespread effect might be associated with the activation of these mechanisms [47]. The analysis showed that the T6 site, measured adjacent to the T6 spinous process, had a greater % change (25%) than the other sites, and thus the greatest hypoalgesic effect, after the high force intervention. A greater local analgesic effect has also been demonstrated in other studies using PPTs and supporting a widespread response [23,29,30]. The trend towards a greater local analgesia might suggest an involvement of both spinal and supraspinal analgesic mechanisms.

This study provides support that PA thoracic mobilizations induce a hypoalgesic effect. The statistical analysis revealed that the effect of time was significant, suggesting that thoracic mobilizations using either high or low force caused significant changes in PPTs.

### Limitations

Τhe aim of this study was to investigate the difference between high and low treatment force and thus it did not include a control group, and the overall treatment effect could be due to factors such as regression to the mean and the non-specific effects of treatment. The hypoalgesic effects of mobilization using a placebo controlled study design have been demonstrated previously, and have been the subject of two recent systematic reviews [9,46].

Another limitation of this study is that only asymptomatic participants were recruited, and it would be beneficial to establish whether the effects of different mobilization forces are observed in symptomatic participants. This would also enable the inclusion of patient reported pain measures. Despite the fact that PPTs have been used in order to investigate the hypoalgesic effect of mobilizations, the clinical relevance of the increased or decreased PPTs still remains unclear [20,21,22,23,25]. A dissociation between PPT values and verbal rating pain has been reported following cervical mobilizations on participants with neck pain [22] and in participants with low back pain [35]. Therefore, future studies using symptomatic participants are necessary for the clear understanding of the clinical relevance of PPT.

Many of the participants in this study were physiotherapy students, so they were not naïve to the potential effects of mobilizations. Therefore, the effects could have been influenced by their expectations [48]. 

## 5. Conclusions

This study demonstrated that high force thoracic PA mobilizations on asymptomatic volunteers elicited a significantly greater immediate hypoalgesic response than low force thoracic mobilizations. Furthermore, this study supported the evidence that mobilizations may induce a widespread hypoalgesic effect as measured by PPTs. Although more studies using symptomatic participants are needed, the results of this study suggest that, where pain and patients’ beliefs allow, clinicians might consider using higher force mobilization treatment. 

## Figures and Tables

**Figure 1 diagnostics-13-00544-f001:**
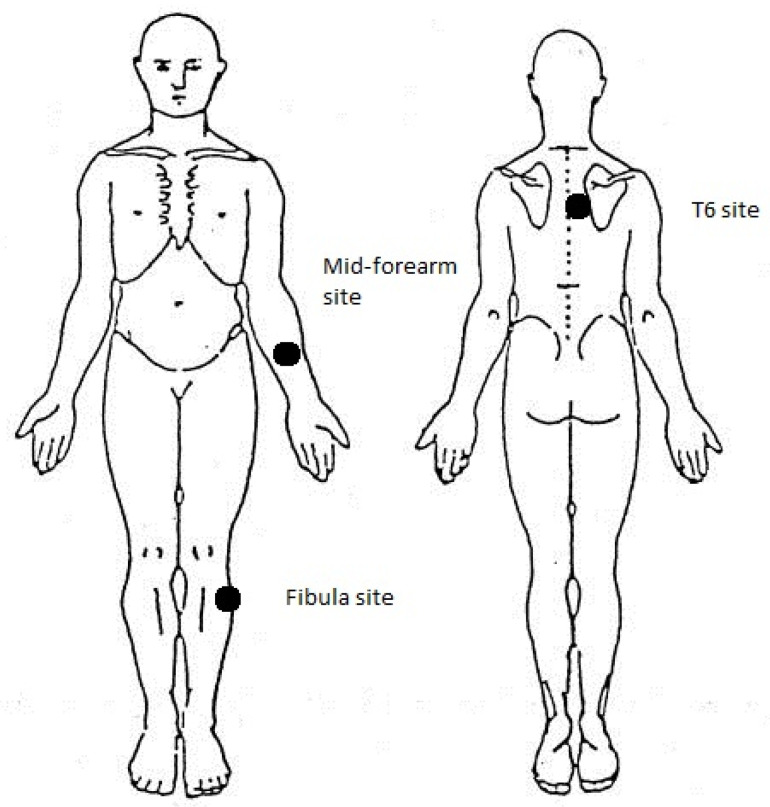
PPT measurement sites: T6 site (paravertebral 3 cm to the right of the T6 spinous process), Mid-forearm site (midway between left wrist and elbow over the anterior ulna), Fibula (2 cm below the left head of fibula).

**Figure 2 diagnostics-13-00544-f002:**
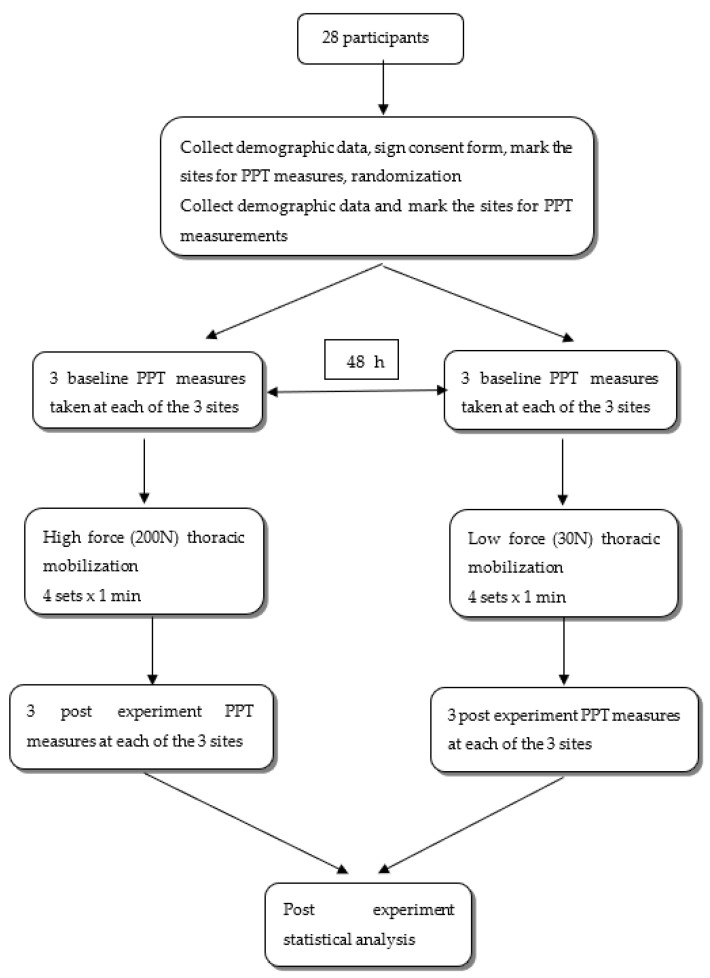
The experimental procedure.

**Figure 3 diagnostics-13-00544-f003:**
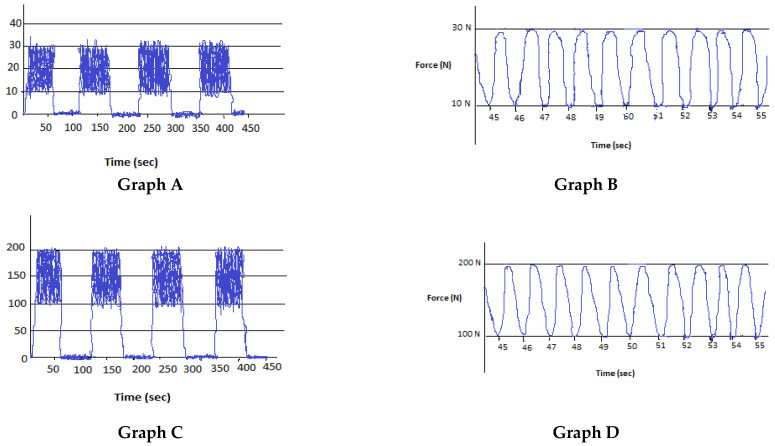
Typical platform recordings for the experimental conditions. **Graph A**: Four sets of low force mobilization. **Graph B:** Recording of low force oscillations from 45 s to 55 s. **Graph C:** Four sets of high force mobilization. **Graph D:** Recording of high force oscillations from 45 s to 55 s.

**Figure 4 diagnostics-13-00544-f004:**
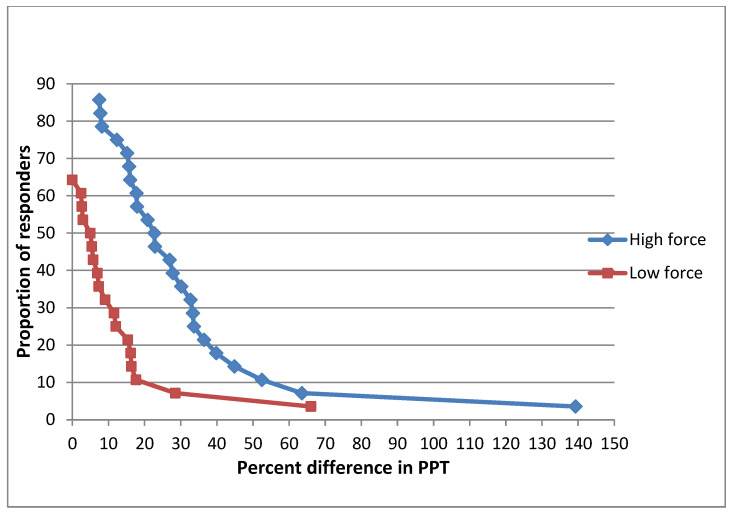
Cumulative proportion of responders analysis graph T6 after high and low mobilizations.

**Figure 5 diagnostics-13-00544-f005:**
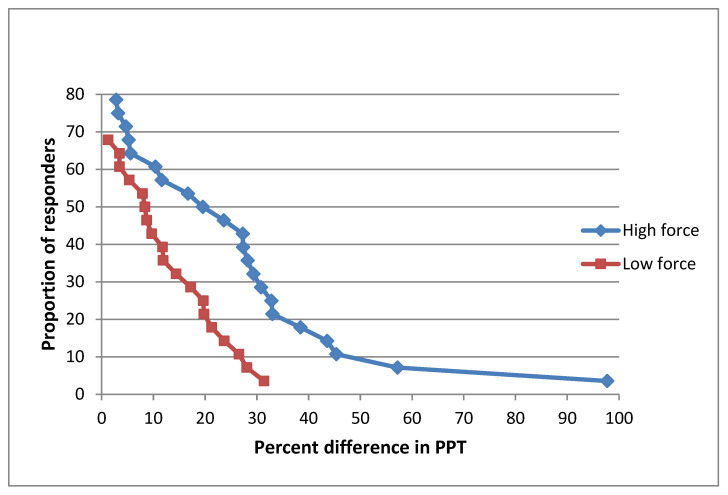
Cumulative proportion of responders analysis graph Mid Forearm after high and low mobilizations.

**Figure 6 diagnostics-13-00544-f006:**
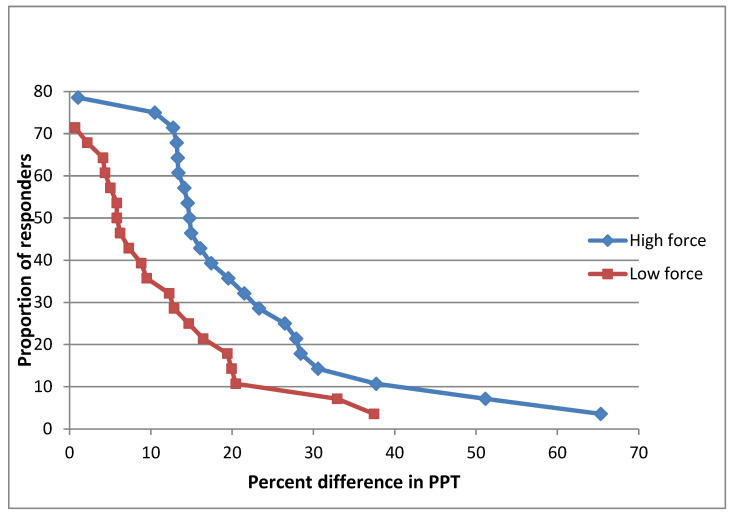
Cumulative proportion of responders analysis graph Fibula after high and low mobilizations.

**Figure 7 diagnostics-13-00544-f007:**
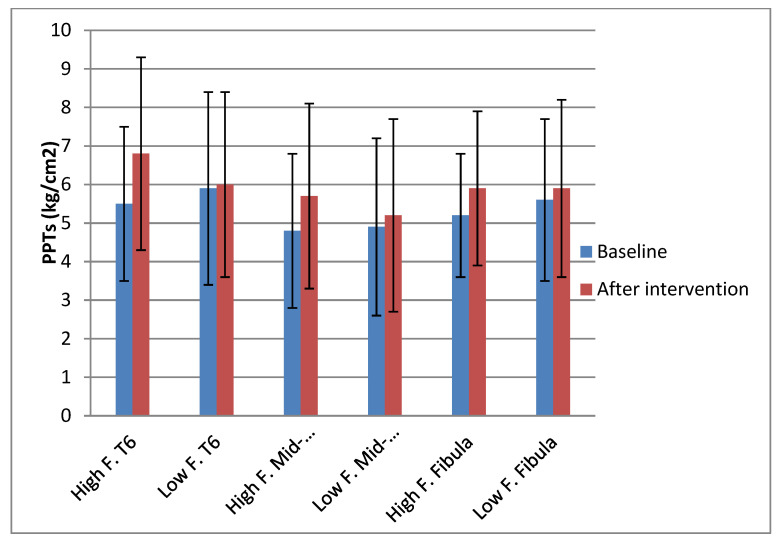
Changes in PPT at each site after high and low force interventions.

**Figure 8 diagnostics-13-00544-f008:**
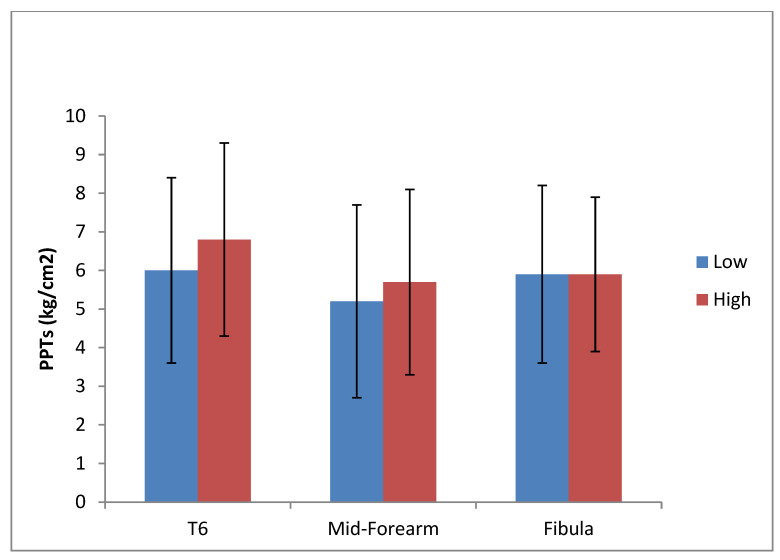
Actual change in PPTs.

**Table 1 diagnostics-13-00544-t001:** $X^{2}$ = Chi-square analysis based on standard error of measurement (SEM). Number of responders (*n* = 28).

	High Force	Low Force	*p* Value
T6 level	14 responders	5 responders	*p* = 0.011
	14 non-responders	23 non-responders	
Mid-forearm	19 responders	6 responders	*p* = 0.000
	9 non-responders	22 non-responders	
Fibula	11 responders	4 responders	*p* = 0.035
	17 non-responders	24 non-responders	

**Table 2 diagnostics-13-00544-t002:** $X^{2}$ = Chi-square analysis based on minimal detectable change (MDC). Number of responders (*n* = 28).

	High Force	Low Force	*p* Value
T6 level	2 responders	0 responders	*p* = 0.150
	26 non-responders	28 non-responders	
Mid-forearm	4 responders	0 responders	*p* = 0.038
	24 non-responders	28 non-responders	
Fibula	0 responders	0 responders	-
	28 non-responders	28 non-responders	

**Table 3 diagnostics-13-00544-t003:** Reliability of baseline measurements.

Site	ICC	95% CI	SEM	MDC
T6 level	0.76	0.54–0.88	1.12	3.1
Mid forearm	0.87	0.75–0.94	0.76	2.1
Fibula	0.74	0.51–0.87	0.94	2.6

ICC = intraclass correlation coefficient, CI = 95% confidence interval, SEM = standard error of measurement, MDC = minimal detectable change.

**Table 4 diagnostics-13-00544-t004:** Mean baseline values, mean increase and percentage change (%) of PPT in each experimental condition. SD = Standard deviation.

Site	High Force			Low Force		
	Mean (SD) Baseline Value (kg/cm^2^) (Range)	Mean (SD) Actual Change (kg/cm^2^) (Range)	%Change (SD) (Range)	Mean (SD) Baseline Value (kg/cm^2^) (Range)	Mean (SD) Actual Change (kg/cm^2^) (Range)	%Change (SD) (Range)
T6 level	5.5 (±2) (11–3.1)	1.2 (±1.3) (4.8–−1.3)	25 (±29.5) (139.3–−23.8)	5.9 (±2.5) (12.4–2.8)	0.2 (±0.9) (2.4–−1.5)	4.4 (±16.8) (66–−16.9)
Mid forearm	4.8 (±2) (9.5–1.8)	0.9 (±0.9) (2.6–−0.6)	19.7 (±24) (97.7–−11.9)	4.9 (±2.3) (11.1–1.4)	0.3 (±0.6) (1.5–−0.9)	6.8 (±14.1) (31.4–−20.2)
Fibula	5.2 (±1.6) (8.5–2.4)	0.7 (±0.9) (2.4–−1)	14.6 (±19.5) (65.3–−18.9)	5.6 (±2.1) (10.5–2.4)	0.3 (±0.7) (1.8–−1)	6.2 (±13.3) (37.5–−20.3)

## Data Availability

Not applicable.

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
