# Peer review of "The Immediate Hypoalgesic Effect of Low and High Force Thoracic Mobilizations in Asymptomatic Subjects as Measured by Pain Pressure Thresholds (PPT)"

_diagnostics, 2023, doi:10.3390/diagnostics13030544_

Round 1
Reviewer 1 Report
The authors present a well-constructed study demonstrating the improved reduction of pain threshold through mobilisation in the thoracic spine in healthy patients. The design is scientifically robust and comprehensible.
In principle, the main points of criticism are recognised by the authors themselves. From my point of view, the study design with physiotherapists as asymptomatic patients and a missing control group without real mobilisation are to be seen as the most important points of criticism. However, this is sufficiently acknowledged in the discussion.
I would only recommend the following 2 minimal adjustments:
1. materials and methods: please consider describing earlier the detailed description of how the pain threshold was measured. In the current form it is not clear for a long time how this was done and leads to some confusion when reading.
2. There is redundancy in the discussion of lines 352-353 and lines 387-388. These are unnecessary repetitions.
Apart from these minimal formal comments, I recommend publication in this form.
Author Response
Point 1: materials and methods: please consider describing earlier the detailed description of how the pain threshold was measured. In the current form it is not clear for a long time how this was done and leads to some confusion when reading.
Response 1: Thank you for your comment. Please know that the PPT measurement info has been placed earlier in the methods section
Point 2: There is redundancy in the discussion of lines 352-353 and lines 387-388. These are unnecessary repetitions.
Response 2: This has been ammended on the manuscript.
Thank you for your valuable suggestions
Reviewer 2 Report
DIAGNOSTICS-2145943 presents findings for force thoracic mobilizations with PPT. While some parts of this paper were interesting, other areas could be improved. I hope the authors consider my feedback.
MAJOR COMMENTS
· Line 99: More details about the power analysis are needed and a citation should not be referred to for the reader. What software was used for the power analysis? What was alpha and beta? What exactly was the power analysis and what data were used for the analysis?
· Section 2.4: Despite the use of a power analysis, which is currently lacking detail, a three-way ANOVA with the current n= is problematic for statistical errors. The same comment somewhat generalizes to the correlation analyses. Perhaps instead the study should be framed as pilot with relevant pilot-level language in the text?
· Table 1 only lists p-values?? What about relevant statistics for comparisons?
· Table 2: Just put the 95% CI in a single column. The abbreviations listed in the table title need to be moved to a note under the table.
· Figure 3 needs to be concluded as it is currently challenging to read. There appears to be spaces between figures.
· Titles and figure legends in Figures 4-8 should be removed and relocated to the figure note, respectively.
MINOR COMMENTS
· Introduction: Some paragraphs in this section could be merged. While this reviewer appreciated the smaller, but larger quantities of paragraphs for readability, this section seems longer than it actually is at this time.
· Line 48: Please be clearer by avoiding “it is hypothesized”. Consider rephrasing.
· Line 101: Units needed for age and BMI.
· Figure 2 could be improved with the use of color.
· Line 334 and elsewhere: Remove the F-value because it informed p-values. Include a relevant comparison difference instead (e.g., mean +- SD). P=0.000 is also not possible and should instead be p<0.001.
· Lines 348-349: Remove results from the Discussion (p=0.000) here and elsewhere.
· Abstract: The sentences before the purpose statement could be reduced or outright deleted to make room for other more important study information like methods and results.
· Make any changes to the abstract that align with those made in the text.
Author Response
Response to Reviewer 2 Comments
Point 1: Line 99: More details about the power analysis are needed and a citation should not be referred to for the reader. What software was used for the power analysis? What was alpha and beta? What exactly was the power analysis and what data were used for the analysis?
Response 1: We used G-Power for the sample size calculation. An a-priori sample size calculation with an effect size of 0.21 an α and β of 0.05 and a power of 0.95, determined that 25 participants would be required. We included 28 in the eventuality of dropouts.
Please know that further information on power analysis details will now be included in the manuscript for your consideration.
Point 2: Section 2.4: Despite the use of a power analysis, which is currently lacking detail, a three-way ANOVA with the current n= is problematic for statistical errors. The same comment somewhat generalizes to the correlation analyses. Perhaps instead the study should be framed as pilot with relevant pilot-level language in the text
Response 2: We used the three way as we assessed condition, time and site. As this study was done in a asymptomatic population, it could be considered as a pilot study. This will now be depicted in the manuscript
Point 3: Table 1 only lists p-values?? What about relevant statistics for comparisons?
Response 3: This table depicts the chi-square analysis of which participants experienced changes in PPT values higher than SEM and MDC (responders) or not (non-responders). Please know that this table has been split in two depicting SEM and MDC separately. More information are now included on either table
Point 4: Table 2: Just put the 95% CI in a single column. The abbreviations listed in the table title need to be moved to a note under the table.
Response 4: Please know that this has now been ammended on the table. New table name is Table 3
Point 5: Figure 3 needs to be concluded as it is currently challenging to read. There appears to be spaces between figures.
Response 5: This has now been ammended
Point 6: Titles and figure legends in Figures 4-8 should be removed and relocated to the figure note, respectively.
Response 6: this has been ammended
Point 7: Introduction: Some paragraphs in this section could be merged. While this reviewer appreciated the smaller, but larger quantities of paragraphs for readability, this section seems longer than it actually is at this time.
Response 7: This has been ammended on the manuscript
Point 8: Line 48: Please be clearer by avoiding “it is hypothesized”. Consider rephrasing.
Response 8: This has been ammended on the manuscript
Point 9: Line 101: Units needed for age and BMI.
Response 9: This has been ammended on the manuscript
Point 10: Figure 2 could be improved with the use of color.
Response 10: Thank you for your comment. Please know that we will keep it as it is due to choice
Point 11: Line 334 and elsewhere: Remove the F-value because it informed p-values. Include a relevant comparison difference instead (e.g., mean +- SD). P=0.000 is also not possible and should instead be p<0.001.
Response 11: This has been ammended in the manuscript
Point 12: Lines 348-349: Remove results from the Discussion (p=0.000) here and elsewhere.
Response 12: This has been ammended in the manuscript
Point 13: Abstract: The sentences before the purpose statement could be reduced or outright deleted to make room for other more important study information like methods and results.
Response 13: This has been ammended in the manuscript
Point 14: Make any changes to the abstract that align with those made in the text.
Response 14: Have gone through the mansuscript and ammended where needed
Thank you for all your valuable and contructive comments
Round 2
Reviewer 2 Report
The authors have done a nice job addressing my previous concerns. One last note is that the authors need to review the order of data elements. For example, Table 1 is first introduced, and then Table 3.
Author Response
Response to Reviewer 2 Comments
Point 1: The authors have done a nice job addressing my previous concerns. One last note is that the authors need to review the order of data elements. For example, Table 1 is first introduced, and then Table 3.
Response 1: Thank you for your comment. Please know that this has now been ammended on the manuscript